# FDA and EMA Oversight of Disruptive Science on Application of Finite Absorption Time (F.A.T.) Concept in Oral Drug Absorption: Time for Scientific and Regulatory Changes

**DOI:** 10.3390/pharmaceutics16111435

**Published:** 2024-11-11

**Authors:** Elias Toulitsis, Athanasios A. Tsekouras, Panos Macheras

**Affiliations:** 1Faculty of Pharmacy, National and Kapodistrian University of Athens, 15784 Athens, Greece; iliastoul@pharm.uoa.gr; 2Department of Chemistry, National and Kapodistrian University of Athens, 15784 Athens, Greece; thanost@chem.uoa.gr; 3PharmaInformatics Unit, ATHENA Research Center, 15125 Athens, Greece

**Keywords:** finite absorption time, oral administration, fraction absorbed, bioavailability, levonorgestrel, amlodipine, theophylline, theotrim, theodur, ketoprofen

## Abstract

**Background:** It has been demonstrated that the concept of infinite absorption time, associated with the absorption rate constant, which drives a drug’s gastrointestinal absorption rate, is not physiologically sound. The recent analysis of oral drug absorption data based on the finite absorption time (F.A.T.) concept and the relevant physiologically based finite-time pharmacokinetic (PBFTPK) models developed provided a better physiologically sound description of oral drug absorption. **Methods:** In this study, we re-analyzed, using PBFTPK models, seven data sets of ketoprofen, amplodipine, theophylline (three formulations), and two formulations (reference, test) from a levonorgestrel bioequivalence study. Equations for one-compartment-model drugs, for the estimation of fraction of dose absorbed or the bioavailable fraction exclusively from oral data, were developed. **Results:** In all cases, meaningful estimates for (i) the number of absorption stages, namely, one for ketoprofen and the levonorgestrel formulations, two for amlodipine, the immediate-release theophylline formulation, and the extended-release Theotrim formulation, and three for the extended-release Theodur formulation, (ii) the duration of each absorption stage and the corresponding drug input rate, and (iii) the total duration of drug absorption, which ranged from 0.75 h (ketoprofen) to 11.6 h for Theodur were derived. Estimates for the bioavailable fraction of ketoprofen and two theophylline formulations exhibiting one-compartment-model kinetics were derived. **Conclusions:** This study provides insights into the detailed characteristics of oral drug absorption. The use of PBFTPK models in drug absorption analysis can be leveraged as a computational framework to discontinue the perpetuation of the mathematical fallacy of classical pharmacokinetic analysis based on the absorption rate constant as well as in the physiologically based pharmacokinetic (PBPK) studies and pharmacometrics. The present study is an additional piece of evidence for the scientific and regulatory changes required to be implemented by the regulatory agencies in the not-too-distant future.

## 1. Introduction


*The health and well-being of the American public depend on FDA’s science-based regulatory decisions.*

*Anonymous*


The term pharmacokinetics was first introduced by F. H. Dost in 1953 in his text, *Der Blutspiegel. Kinetik der Konzentrationsabläufe in der Kreislaufflüssigkeit* [1]. A revised edition of this book entitled *Grundlagen der Pharmacokinetik* was published in 1968 [2]. In this latter book, Dost’s law of corresponding areas was stated as follows: “the ratio of the area beneath the blood level-time curves, after oral administration to that following intravenous administration of the same dose, is a measure of the absorption of the drug administered” [2,3]. Few years later in 1976, the late John Wagner suggested [3,4] that Dost’s law of corresponding areas be replaced by Equation (1), (1)$$FF*=\frac{{\left[\int_{0}^{\infty }\mathrm{Cdt}\right]}_{\mathrm{po}}}{\int_{0}^{\infty }\mathrm{Cdt}}\frac{D_{\mathrm{po}}}{D}$$
where *F** is the fraction of the dose, *D_po_*, which is absorbed (0 < 1), *F* is the bioavailable fraction due to the first-pass effect (0 < *F* < 1), *D* is the intravenous dose, and the integrals are the total areas under the concentration–time curves following oral and intravenous administrations, respectively; he also showed that, in some cases, *F* = 1, and in others, *F* ≠ 1.

In these first two books of pharmacokinetics [1,2], Dost used Equation (2) to describe the blood concentration *C* as a function of time *t* after the oral administration of dose *D_po_*, assuming one-compartment-model disposition and first-order absorption and elimination, (2)$$C=\frac{FD_{\mathrm{po}}k_{a}}{V_{d}(k_{a}-k_{\mathrm{el}})}(e^{-k_{\mathrm{el}}t}-e^{-k_{a}t})$$where *V_d_* is the volume of distribution, *k*_a_ is the first-order rate constant of absorption, and *k_el_* is the elimination first-order rate constant. Since both rate constants, *k_a_* and *k_el_*, are first-order, Equation (2) implies that both absorption and elimination run for infinite time. In fact, Dost utilized the function developed by H. Bateman [5] back in 1910 for the decay of the radioactive isotopes to describe oral drug absorption as a first-order process; see Equation (2). This false hypothesis implies infinite drug absorption time and is against physiology and common wisdom, i.e., drugs are absorbed in finite time.

Until the late 1960s, variability in drug response was always associated with the patient in accord with Sir William Osler’s variability principle, i.e., “Variability is the law of life, and as no two faces are the same, so no two bodies are alike, and no individuals react alike and behave alike under the abnormal conditions which we know as disease” [6,7]. However, in the late 1960s, it was realized that a variable or poor response to a therapeutic agent may not have its origin in the patient; it may be due to a formulation defect in the drug product administered, the so called “bioavailability problem” [8,9]. Therefore, the adoption of the bioavailability concept by FDA [10] in 1977 is the logical consequence of the relevant in vitro and in vivo experimental observations published in late 1960s and early 1970s [8,9]. According to FDA definition “**Bioavailability is the rate and extent to which the active ingredient or active moiety is absorbed from a drug product and becomes available at the site of action**”. For regulatory purposes, FDA used the area under the curve (AUC) as an extent metric in accord with Equation (1) and the maximum blood concentration *C*_max_ observed as a rate metric; mathematically, *C*_max_ can be obtained by equating the first derivative of Equation (2) with zero. In parallel, Equation (2) has been used in pharmacokinetics since 1953 for fitting purposes; since then, the absorption rate constant, *k*_a_, is the sole parameter utilized to quantify the drug input rate until today.

However, the unphysical hypothesis of infinite time for oral drug absorption associated with Equation (2) was proposed in 2019 [11]. A disruptive paper introducing the overlooked-for-decades physiologically sound finite absorption time (F.A.T.) concept in oral drug absorption was published in 2020 (see Figure 2 in [12]). Specific time constraints for drug absorption from the small intestines and colon, at 5 h and 30 h, respectively, are applied in accord with the literature values [13]. The high blood flow rate in the portal vein (20–40 cm/s) [14] imposes the rapid removal of the absorbed drug molecules towards the liver maintaining sink conditions in the first-order drug transfer; thus, oral drug absorption obeys zero-order kinetics; see Figure 1. Furthermore, the relevant physiologically based finite-time pharmacokinetic (PBFTPK) models were developed and used for the analysis of oral concentration and time data [15]. Using the PBFTPK models, the kinetics of drug absorption is described in terms of the number of absorption stages and their corresponding input rate, which replace the fallacious unique first-order absorption rate constant [15,16,17,18]. In parallel, the bioavailability parameters AUC and *C*_max_ were re-interpreted [19,20], novel bioequivalence metrics for the rate and extent of absorption were suggested [21], and the construction of (i) the percent absorbed versus time curves and (ii) the in vitro–in vivo correlations were revamped [22,23], while the finite dissolution time was introduced and used in the realm of BCS [24].

The 21st century is witnessing the incorporation of advanced quantitative methods into regulatory science via the FDA’s Model-Informed Drug Development Initiative (MIDD) that was formally announced in 2018 [25,26,27,28]. MIDD makes the drug development process more rational and efficient by integrating data from mathematical and statistical models predicting drug’s effects and reducing unnecessary patient exposure. Two FDA guidances, i.e., “Population pharmacokinetics, guidance for industry, 3 February 2022” [29] and “The use of physiologically based pharmacokinetic analyses-biopharmaceutics applications for oral drug product development, manufacturing changes and controls, Guidance for industry, 30 September 2020” [30], assist FDA applicants in the application of population pharmacokinetic analysis and the development of physiologically based pharmacokinetic (PBPK) models, respectively. In neither of these guidances, the PBFTPK models are mentioned. Thus, the classical first-order absorption kinetic models are always applied in classical pharmacokinetics, in PBPK studies, and population analyses since the absorption rate constant is wrongly considered as a pivotal parameter of oral drug absorption [17,31]. It should be mentioned that a talk with the provocative title “*The estimates for the absorption rate constant in pharmacokinetics and pharmacometrics are wrong: A new era based on the finite absorption time concept rises*” was presented and very well received at the prestigious PAGE meeting in A Coruña, Spain [32].

In this study, we re-analyze and re-interpret oral studies [33,34,35,36] using PBFTPK models. The most recent study [36] focuses on the development of convolution approaches in compartmental pharmacokinetic models and application to non-bioequivalent formulations; in this study, our F.A.T. research is quoted, but not used for the analysis of data. Herein, we demonstrate the utility of PBFTPK models for the analysis of oral data published in [33,34,35,36]; the PBFTPK models not only reveal the detailed characteristics of drug absorption, but also provide meaningful parameter estimates for each one of the consecutive absorption stages. Furthermore, estimates for the absolute bioavailability or fraction of dose absorbed for drugs obeying one-compartment model disposition are derived based on oral data exclusively.

## 2. Methods

The concentration and time data of the plots in [33,34,35,36] were digitized by transferring the published figures to the Windows utility MS Paint, reading off the coordinates of axis ranges and data points, and performing linear interpolation to recover the data shown in the published papers. They were then analyzed with a variety of models assuming one- and two-compartment disposition. The least-squares method was implemented within the programming environment of Igor Pro 9 by WaveMetrics [15,29] for all fitting studies. Parameter uncertainties, co-variances, and correlations between them were determined to help assess the quality of each fit. Fit residuals, i.e., differences between experimental and calculated points, were also plotted as an additional criterion for the quality of each fit.

The estimates for absolute bioavailability, *F*, for the one-compartment-model drugs with one zero-order input stage were derived from Equation (3) [19].
(3)$$F=\frac{k_{el}\tau }{e^{k_{el}\tau }-1}$$
where *k_el_* is the elimination rate constant, and *τ* is the duration of drug absorption.

The estimate for *F* for the one-compartment-model drugs with two input stages was derived from Equation (4) (see Appendix B),(4)$$F=(\frac{F_{1}D}{V_{d}}+\frac{F_{2}D}{V_{d}})k_{el}{\left[\frac{F_{1}D}{V_{d}\tau_{1}}(e^{k_{el}\tau_{1}}-1)+\frac{F_{2}D}{V_{d}\tau_{2}}e^{k_{el}\tau_{1}}(e^{k_{el}\tau_{2}}-1)\right]}^{-1}$$
where *D* is the dose, *V_d_* is the volume of distribution, and *F*_1_ and *F*_2_ are the bioavailable fractions of the stages 1 and 2 with duration *τ*_1_ and *τ*_2_, respectively. The estimates for $\frac{F_{1}D}{V_{d}}$ and $\frac{F_{2}D}{V_{d}}$ and all other parameters of Equation (4) are derived from the fitting of the corresponding PBFTPK model to the experimental data.

A semi-noncompartmental approach developed in [19] and based on Equation (5) was also used for the estimation of the fraction of the dose absorbed *F** for drugs obeying one-compartment-model disposition with one, two, or three zero-order absorption stages. (5)F*=([AUC]0∞oral[AUC]0∞hy.i.v.)12
where ${{\left[AUC\right]}_{0}^{\infty }}_{oral}$ is the area under the concentration time curve of the oral formulation and ${{[\left(AUC\right)}_{0}^{\infty }]}_{hy.i.v.}$ corresponds to the area of the hypothetical intravenous bolus administration of the same dose derived from the back extrapolation of the elimination phase experimental oral data beyond time *τ* of the oral dose. The use of Equation (5) does not involve a separate intravenous drug administration of the same dose to determine the corresponding area under the curve. Therefore, the use of Equation (5) provides an estimate for the fraction of dose absorbed *F** and not the bioavailable fraction *F* since the value of the parameter ${{\left[AUC\right]}_{0}^{\infty }}_{hy.i.v}$ is affected by the first-pass effect, if any. However, when first-pass effect is not operating, *F = F**.

It should be noted that the estimates for *F* and *F** derived from Equations (3)–(5) are based on compartmental approaches and therefore are subject to the error associated with the quality of fitting of the PBFTPK models to experimental data.

## 3. Results

All observed data sets reported in [33,34,35,36] were analyzed using nonlinear regression analysis based on PBFTPK models [15]. All data sets were described nicely by the PBFTPK models; see Figure 2, Figure 3 and Figure 4. The fitting results using the Bateman equation (Equation (2)) to all data sets were inferior as shown in the Appendix A. The findings presented in Figure 2, Figure 3 and Figure 4 are tabulated in Table 1, which shows the estimates for the duration of drug absorption stages and the drug input rate for each one of the input stages. Ketoprofen (Figure 2A) and levonorgestrel (reference and test) (Figure 4) exhibit a single input stage with 0.75 and ~1 h, respectively, while amlodipine (Figure 2B) has two input stages with 2.1 and 5.3 h durations, respectively. These values indicate that the absorption of ketoprofen and levonorgestrel (reference and test) is completed in the small intestines, while amlodipine is also absorbed in the colon. The three theophylline formulations exhibit different absorption profiles. The immediate-release formulation of theophylline (Figure 3) exhibits a very high initial input rate (0.0068 mg h/mL) of 1.2 h duration followed by a much lower input rate (0.0011 mg h/mL) for 1.3 h; this means that the entire absorption process lasts for 2.5 h, which indicates that theophylline absorption from the immediate-release formulation terminates in the small intestines; see Table 1. The absorption of theophylline from the extended-release formulation Theotrim exhibits a very slow initial input rate (0.0018 mg h/mL) for a short period of time (~0.7 h) followed by a second stage of absorption, which lasts for 6.3 h with even lower input rate (0.0007 mg h/mL); see Table 1. The total absorption duration of theophylline from Theotrim is 7 h, which is indicative of the extended-release character of this formulation since absorption of theophylline continues to take place in the upper part of the colon. The absorption of theophylline from Theodur exhibits three successive stages, with a total duration of absorption close to 11.6 h; see Table 1. The initial stage with slow input rate (0.0006 mg h/mL) lasts 4.2 h and is followed by an increase in theophylline’s rate of absorption (0.001 mg h/mL); during the last third stage of absorption of 4.2 h, the rate is diminished to 0.0004 mg h/mL. These results show that a significant portion of theophylline absorption from Theodur takes place in the colon. Overall, the PBFTPK models provide a detailed picture of the absorption process of theophylline from the three formulations. For comparative purposes, Figure 5 shows the poor fitting of Equation (2) (Bateman function) to Theodur data; the uncertainty of the parameter estimates makes them fully unreliable. Figure 4 shows the similarity of the parameter estimates derived from the fitting of the PBFTPK model with one input stage and two compartment model disposition to reference and test formulation of levonorgestrel. The durations of levonorgestrel absorption, *τ,* for the test, i.e., 0.96 ± 0.06 h^−1^, and for the reference formulation, i.e., 1.09 ± 0.08 h^−1^, are remarkably similar. This observation underlines the small variability of *τ* for the two levonorgestrel formulations studied.

We also constructed the fraction absorbed versus time plots for the three formulations of theophylline following the methodology based on the F.A.T. [22] and compared the plots with the corresponding plots reported in the literature [36]; see Figure 6, Figure 7 and Figure 8. For the immediate-release theophylline formulation, our approach (Figure 6A) shows that absorption reaches the theoretical maximum (fraction absorbed = 1) at 1.7 h. The reported plot (Figure 6B) shows an asymptotic increase in fraction absorbed reaching a plateau (0.8) at 3 h. This is unexplained since both the classical Wagner–Nelson method and our approach based on F.A.T., by definition, always reach fraction absorbed = 1 [22]. The Theotrim results are shown in Figure 7. Again, the plot based on the F.A.T. in Figure 7A [22] shows that the expected maximum (fraction absorbed = 1) is reached in 7.4 h, which is very close to the estimate 7.0 h derived from the nonlinear regression analysis fit (Figure 3B and Table 1). On the contrary, the plot reported in [36] and replotted in Figure 7B exhibits an increase in the fraction absorbed as a function of time for 14 h, reaching asymptotically 0.8, which is remarkably lower than the expected maximum fraction absorbed = 1. The Theodur plots shown in Figure 8 have similar profiles with Theotrim. The plot, Figure 8A, derived from the F.A.T. analysis [22] shows the three successive absorption phases (Figure 3C and Table 1) reach the expected plateau maximum at 10.5 h, which is close to the *τ* estimate 11.6 h quoted in Table 1; the reported plot [36] for Theodur shows an asymptotic increase in the fraction absorbed for 14 h, reaching a questionable plateau, fraction absorbed = 0.6. All results of the theophylline formulations are also important for the pharmaceutical scientists working on drug dissolution. In fact, the current dissolution tests have limited predictability since the zero-order drug input is not taken into consideration. A two-phase system based on drug dissolution–uptake is most akin to in vivo conditions since it provides a zero-order input as observed in our previous study [22].

The estimates of the bioavailable fraction for the drugs following one-compartment-model kinetics are reported in Table 2. The estimate for ketoprofen of 0.98 underlines its complete absorption and not only agrees with the literature data [37] but also indirectly verifies the absence of first-pass effect. The lower estimate of 0.78 for the bioavailable fraction, given in Table 2, should be linked with the prevailing role of the value of *τ* in the estimation of *F* from oral data using Equation (3), i.e., the error associated with the estimation of *τ* is propagated to the estimate for *F*. The estimate (0.91), Table 2, derived from Equations (4) and (5) for theophylline’s bioavailable fraction corresponding to the immediate-release formulation, agrees with the literature data [38,39]. The estimates (0.8, 0.84, and 0.75) were derived for *F* from the analysis of the extended-release Theotrim and Theodur theophylline formulations, respectively; see Table 2. Although the *F* estimates for Theotrim are not reported in the vast literature of theophylline formulations, Theodur exhibits complete absorption (*F* = 0.97) when administered in doses of 100 to 300 mg [40,41]. Our semi-compartmental estimation approach based exclusively on oral data and the use of Equation (5) is heavily dependent on the estimation of *τ* and leads to a much lower estimate for *F*, i.e., 0.75. Due to the complex nature of theophylline absorption from Theodur, as seen in Figure 3C, the estimation of *τ* is subject to error, which affects the ${{\left[AUC\right]}_{0}^{\infty }}_{hy.i.v}$ parameter in the denominator of Equation (5). In the present case, this results in underestimation of *F*.

## 4. Discussion

This study is an additional piece of evidence for the valid use of the physiologically sound F.A.T. concept in oral drug absorption. According to Figure 1, there is no physical, mathematical, or physiological reason to perpetuate the fallacy of first-order kinetics in oral drug absorption. First-order kinetics should be applied in systems that operate under non-sink conditions. In these systems, reversible first-order transport is observed, and concentration equilibrium between the two sides of the cell membrane can be reached. For example, the Bateman equation has been nicely applied in the transport of Ca^2+^ between cells [42].

By coupling the results of the present study with the scientific and regulatory implications of the F.A.T. concept pointed out previously [16,17,18,19,20,21,22,23,24], a re-consideration of the relevant guidelines [29,30,43,44,45,46] is due. This is particularly so if one recalls the introductory text for the mission of the FDA in relation to the scientific guidelines.

Steps can be taken by the FDA and the EMA to discontinue the perpetuation of the fallacy of first-order kinetics in oral drug absorption during the assessment/approval of the dossier including PBPK and/or pharmacometrics studies of the pharmaceutical companies [29,30]. Although there is a distinct lack of structural and mechanistic modelling in the currently available PBPK and pharmacometrics software to integrate finite-time absorption modelling, the relevant regulatory guidelines [29,30] can be amended and include PBFTPK models for the analysis of oral studies. Subsequently, new avenues for pediatric and interspecies pharmacokinetic scaling will be opened since the current use of the absorption rate constant has no real physical/physiological meaning [47]. In parallel, PBPK studies investigating oral drug absorption in children will benefit if coupled with the top–down PBFTPK models [16]. For example, our study [23] on the analysis of carbamazepine pharmacokinetics in adults using PBFTPK models revealed the very long duration (>16 h) of carbamazepine absorption; this finding is of extreme importance and can be applied and tested using the published PBPK studies focusing on children’s data [48]. A visual inspection of the observed data in [48] reveals the long duration of carbamazepine absorption in children too.

In the same vein, the FDA and EMA guidances dealing with bioequivalence studies or in vitro–in vivo correlations [43,44,45,46] can be reconsidered in the light of the F.A.T.-based concerns published in the last four years [16,17,18,19,20,21,22,23,24] such as the bioequivalence metrics, the duration of sampling in bioequivalence studies, the use of partial areas, and the physiological time constraints for drug dissolution for IVIVC. Finally, it should be emphasized that indirect additional support for the validity of the F.A.T. concept has been provided by machine learning techniques [49,50], which suggest “average slope” to be the most suitable rate metric in bioequivalence studies; this is in full agreement with the zero-order drug input kinetics associated with the sink conditions of the absorption process depicted in Figure 1 and the slope of the amount absorbed versus time curve suggested in [21] for the assessment of rate of absorption in bioequivalence studies.

## 5. Conclusions

This study provides insights into the detailed characteristics of oral drug absorption of seven formulations. The use of PBFTPK models in drug absorption analysis can be leveraged as a computational framework to discontinue the perpetuation of the mathematical fallacy of classical pharmacokinetic analysis based on the absorption rate constant as well as in the physiologically based pharmacokinetic (PBPK) studies and pharmacometrics. The present study is an additional piece of evidence for the scientific and regulatory changes required to be implemented by the regulatory agencies.

## Figures and Tables

**Figure 1 pharmaceutics-16-01435-f001:**
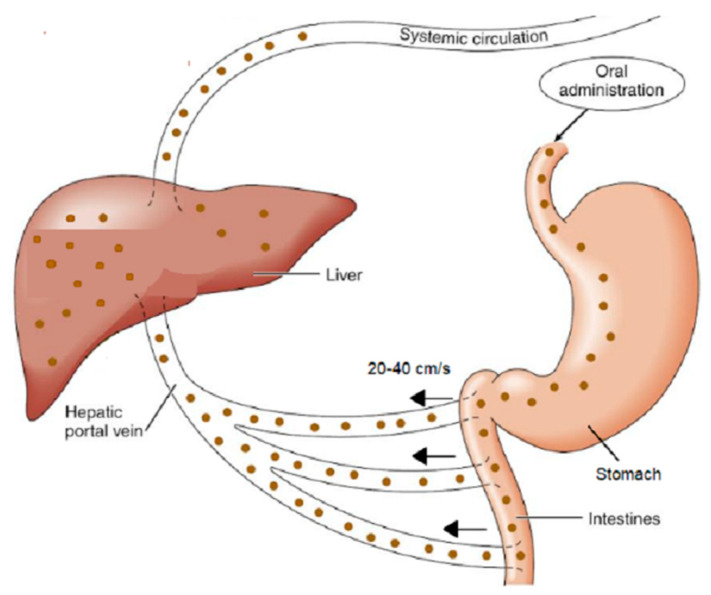
A schematic of the drug molecules (solid circles) moving from the small intestine to the portal vein, where a rapid blood flow (20–40 cm/s) [14] maintains sink conditions throughout the drug absorption process.

**Figure 2 pharmaceutics-16-01435-f002:**
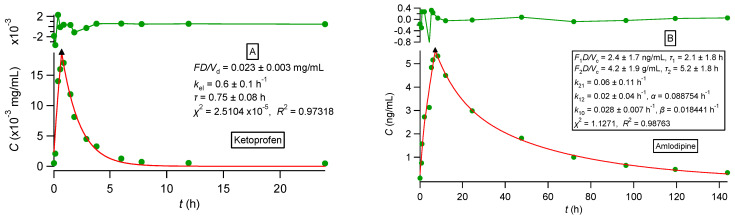
(**A**) Best fit results of Equations (29) and (30) reported in [15] for ketoprofen experimental data [33]. (**B**). Best fit results of Equations (42), (44) and (46) reported in [15] for amlodipine experimental data [34]. The symbol ▲ denotes the end of the absorption process. The top panel depicts the fit residuals.

**Figure 3 pharmaceutics-16-01435-f003:**
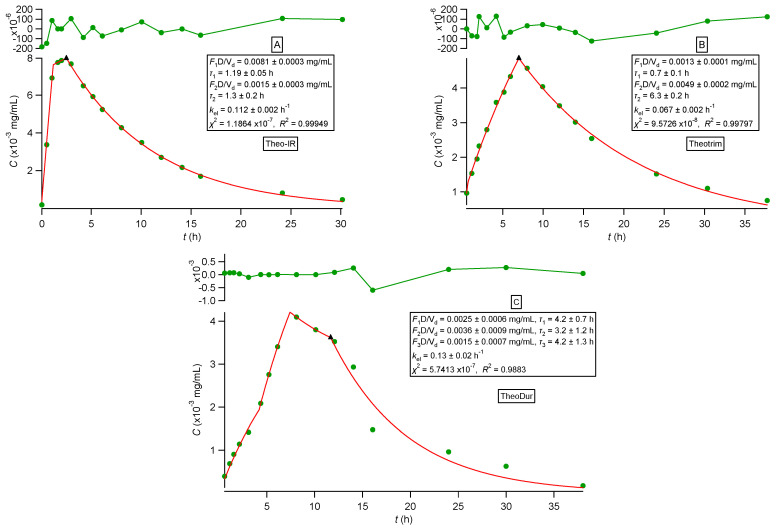
(**A**) Best fit results of Equations (31)–(33) reported in [15] for theophylline (immediate-release) experimental data [35]. (**B**) Best fit results of Equations (31)–(33) reported in [15] for Theotrim (extended-release) experimental data [35]. (**C**) Best fit results of Equations (34)–(37) reported in [15] for Theodur (extended-release) experimental data [35]. The symbol ▲ denotes the end of the absorption process. The top panel depicts the fit residuals.

**Figure 4 pharmaceutics-16-01435-f004:**
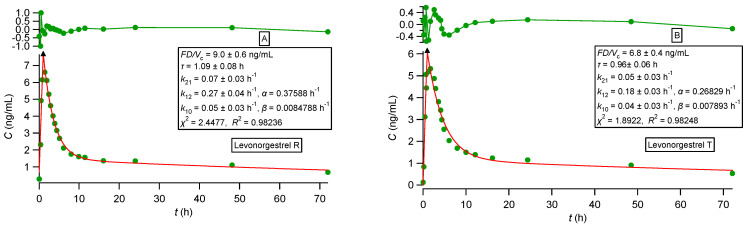
(**A**) Best fit results of Equationss (38) and (40) reported in [15] for levonorgestrel (reference) experimental data [36]. (**B**) Best fit results of Equations (38) and (40) reported in [15] for levonorgestrel (test) experimental data [36]. The symbol ▲ denotes the end of the absorption process. The top panel depicts the fit residuals.

**Figure 5 pharmaceutics-16-01435-f005:**
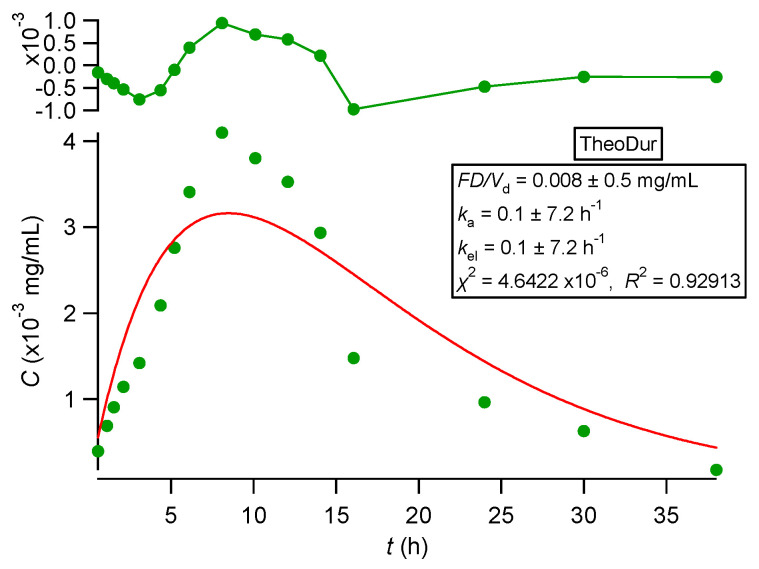
Fitting Equation (2) to Theodur data [35].

**Figure 6 pharmaceutics-16-01435-f006:**
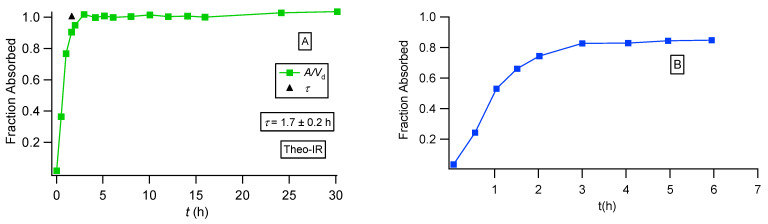
Fraction absorbed versus time plots for the theophylline immediate-release formulation calculated in two different ways: (**A**) based on PBFTPK parameters with symbols defined in [35] and the black triangle denoting the termination of drug absorption and (**B**) data replotted from [36].

**Figure 7 pharmaceutics-16-01435-f007:**
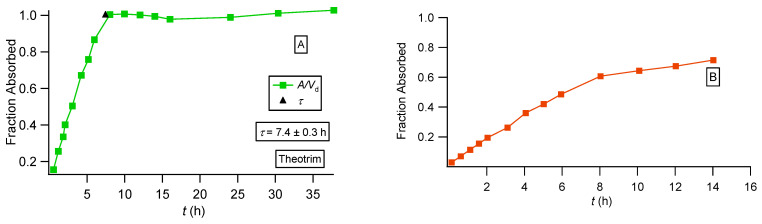
Fraction absorbed versus time plots for the Theotrim formulation calculated in two different ways: (**A**) based on PBFTPK parameters with symbols defined in [35] and the black triangle denoting the termination of drug absorption and (**B**) data replotted from [36].

**Figure 8 pharmaceutics-16-01435-f008:**
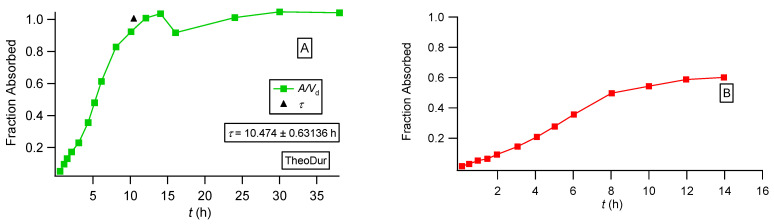
Fraction absorbed versus time plots for the Theodur formulation calculated in two different ways: (**A**) based on PBFTPK parameters with symbols defined in [35] and the black triangle denoting the termination of drug absorption and (**B**) data replotted from [36].

**Table 1 pharmaceutics-16-01435-t001:** Drug absorption estimates derived from fitting of PBTPK models to observed data reported in [33,34,35,36].

Drug	Number of Input Stages	Duration for Each Stage (h)	Total Duration (h)	Input Rate for Each Stage (mg h/mL)
Ketoprofen	1	0.75	0.75	0.031
Amlodipine	2	2.1, 5.3	7.4	1.16, 0.80 *
Immediate-release theophylline	2	1.20, 1.3	2.5	0.0068, 0.0011
Theotrim	2	0.72, 6.28	7.0	0.0018, 0.0007
TheoDur	3	4.2, 3.2, 4.2	11.6	0.0009, 0.0024, 0.0016
Levonorgestrel (reference)	1	1.09	1.09	8.3 *
Levonorgestrel (test)	1	0.96	0.96	7.1 *

* Expressed in ng h/mL.

**Table 2 pharmaceutics-16-01435-t002:** Estimates for *F* of drugs obeying one-compartment-model disposition.

Drug	Equation (3)	Equation (4)	Equation (5)
Ketoprofen	0.78		0.98
Immediate-release theophylline		0.91	0.91
Theotrim		0.8	0.84
TheoDur			0.75

## Data Availability

All data analyzed in this study were found in the published literature as indicated in the references.

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
