# Peer review of "FDA and EMA Oversight of Disruptive Science on Application of Finite Absorption Time (F.A.T.) Concept in Oral Drug Absorption: Time for Scientific and Regulatory Changes"

_pharmaceutics, 2024, doi:10.3390/pharmaceutics16111435_

Round 1

Reviewer 1 Report

Comments and Suggestions for Authors

Thanks for this manuscript. 

A few comments you may want to consider for a revision and further work:

In how far do you consider the transit time variability in the models? 

I think it would be valuable to also plot unsuccessfuly non-FAT modeling approaches in this manuscript to show the superiority of your modeling strategy, similar to what you do with figure 5 so that the use cannot be questioned. In my opinion showing improved accuracy by better prediction error of Cmax and AUC could be a good way.

I understand you do aim to challenge the paradigm of first order absorption kinetics. As this is very substantial and may invalidate a lot of papers, I think it could be helpful for provide further context, also in this manuscript to give your argument more weight. I understand you do reason this by the blood flow rate in the portal vein creating an absorption sink. Readers familar with dissolution testing however might ask: Is my in vitro dissolution test under sink conditions not following first-order kinetic? 

Author Response

Comment 1: In how far do you consider the transit time variability in the models?
Response 1: The transit time variability is not considered in this work. We focus on the variability of the duration of the drug absorption stages. We added a relevant comment regarding the variability of τ estimates for the two formulations of levonorgestrel.

Comment 2: I think it would be valuable to also plot unsuccessfuly non-FAT modeling approaches in this manuscript to show the superiority of your modeling strategy, similar to what you do with figure 5 so that the use cannot be questioned. In my opinion showing improved accuracy by better prediction error of Cmax and AUC could be a good way.
Response 2: We provide the fitting results of the Bateman equation to the seven data sets analyzed in Figs. 1-3. All these results are quoted in the Supplementary material, Figs. S1-S3.

Comment 3: I understand you do aim to challenge the paradigm of first order absorption kinetics. As this is very substantial and may invalidate a lot of papers, I think it could be helpful for provide further context, also in this manuscript to give your argument more weight. I understand you do reason this by the blood flow rate in the portal vein creating an absorption sink. Readers familar with dissolution testing however might ask: Is my in vitro dissolution test under sink conditions not following first-order kinetic?
Response 3: We added some text to address this point in the results section.

Reviewer 2 Report

Comments and Suggestions for Authors

After conducting a thorough review of this paper, it is evident that the authors have multiple prior publications featuring the same data and figures. This includes references to their earlier works, as well as statements regarding the FDA and EMA. If the current paper aims to reanalyze and reinterpret findings from these previous studies, it is essential for the authors to provide a detailed explanation in the introduction. Specifically, they should outline which aspects of their past research are being revisited, the role of other publications, and how they are being re-evaluated. Additionally, the authors should highlight the new insights or contributions that this paper offers. This transparency will help readers understand the context and significance of the current work about the previous research. The authors must justify the use of the same images in this paper as in their previous work.

Author Response

Comment 1: After conducting a thorough review of this paper, it is evident that the authors have multiple prior publications featuring the same data and figures. This includes references to their earlier works, as well as statements regarding the FDA and EMA. If the current paper aims to reanalyze and reinterpret findings from these previous studies, it is essential for the authors to provide a detailed explanation in the introduction. Specifically, they should outline which aspects of their past research are being revisited, the role of other publications, and how they are being re-evaluated. Additionally, the authors should highlight the new insights or contributions that this paper offers. This transparency will help readers understand the context and significance of the current work about the previous research. The authors must justify the use of the same images in this paper as in their previous work.

Response 1: In this work we present for the first time analysis of controlled release formulations using PBFTPK models (see Fig. 3 and Tables 1 and 2). We also developed a new Equation (Eq. 4) for the estimation of absolute bioavailability for drugs exhibiting two input phases and one compartment model disposition. We re-interpreted published % absorbed vs time data and demonstrated their obvious drawbacks (see Figs. 6-8).

Fig 1A was modified from our previous work [13] for the benefit of the reader. Figure 1B had not been published previously. It emphasizes the zero-order input kinetics as a result of the sink conditions maintained in the neighborhood of the G.I. membrane and portal vein.